# Dynamic Transcriptome Analysis Reveals Transcription Factors Involved in the Synthesis of Ethyl Acetate in Aroma-Producing Yeast

**DOI:** 10.3390/genes13122341

**Published:** 2022-12-11

**Authors:** Bingqian Ni, Weiwei Li, Kiren Ifrah, Binghao Du, Youqiang Xu, Chengnan Zhang, Xiuting Li

**Affiliations:** 1Key Laboratory of Brewing Microbiome and Enzymatic Molecular Engineering, China General Chamber of Commerce (BTBU), Beijing 100048, China; 2Beijing Advanced Innovation Center for Food Nutrition and Human Health, Beijing Technology and Business University (BTBU), Beijing 100048, China; 3School of Food and Health, Beijing Technology and Business University (BTBU), Beijing 100048, China

**Keywords:** ethyl acetate, transcription factor, RNA-Seq, aroma-producing yeast

## Abstract

Ethyl acetate is an important flavor element that is a vital component of *baijiu*. To date, the transcription factors that can help identify the molecular mechanisms involved in the synthesis of ethyl acetate have not been studied. In the present study, we sequenced and assembled the *Wickerhamomyces anomalus* strain YF1503 transcriptomes to identify transcription factors. We identified 307 transcription factors in YF1503 using high-throughput RNA sequencing. Some transcription factors, such as C2H2, bHLH, MYB, and bZIP, were up-regulated, and these might play a role in ethyl acetate synthesis. According to the trend of ethyl acetate content, heat map results and STEM, twelve genes were selected for verification of expression levels using quantitative real-time PCR. This dynamic transcriptome analysis presents fundamental information on the transcription factors and pathways that are involved in the synthesis of ethyl acetate in aroma-producing yeast. Of significant interest is the discovery of the roles of various transcription factor genes in the synthesis of ethyl acetate.

## 1. Introduction

*Baijiu* is a traditional Chinese alcoholic beverage that has been produced for more than 2000 years and is very popular [1,2,3,4]. *Baijiu* has numerous flavor components, and contains over 1870 volatile compounds, many of which are esters [5,6,7,8]. The principal characteristics of *baijiu* are its fruity and sweet aroma from esters, and its aroma and flavor are distinct from those of other beverages. Among the esters in *baijiu*, ethyl acetate has been recognized as a major aroma constituent and its content fluctuates among different *baijiu* varieties [6]. Ethyl acetate also contributes to strong and light flavors and the distinct flavors of *Xifeng* and rice *baijiu* [9,10]. Consequently, ethyl acetate is used as a standard for evaluating all varieties of *baijiu* [10,11,12]. 

Ethyl acetate in *baijiu* is mainly produced by metabolism of microbes such as bacteria, mold, and yeast strains during *baijiu* fermentation [13]. The aroma-producing yeasts, which are also called ester-manufacturing yeasts, are a class of wild-type non-saccharomyces yeasts. These yeasts are considered as the principal contributors to production of ethyl acetate [14,15,16,17,18]. Because of their important contribution to the flavor of *baijiu*, aroma-producing yeasts are referred to as “flavor regulators” [19].

In a previous study, we found that *Wickerhamomyces anomalus* strain YF1503 produced a high concentration of ethyl acetate. Under optimum fermentation conditions, strain YF1503 produced 17.35 g/L of ethyl acetate [20]. To date, research on the synthesis of ethyl acetate has mainly focused on determining the genes coding for the enzymes. Previous research has shown that ethyl acetate synthesis in yeast is carried out by alcohol acetyl transferases, hemiacetal dehydrogenases, and esterases [16,21]. Besides these three types of enzymes, there may be other factors involved in the synthesis of ethyl acetate. 

Transcription factors have important roles as master switches in complex regulatory networks, and are essential in plant, animal, and medical research [22,23]. However, the roles of transcription factors in the synthesis of ethyl acetate have not been clarified. To gain a comprehensive understanding of the ethyl acetate pathway and its regulation, we generated a transcriptome map of YF1503. RNA sequencing technology is a comprehensive method for research on dynamic transcriptomes and is useful for understanding and characterizing the transcriptome in model and non-model species [24,25]. The transcriptome denotes almost all the genes in a cell or organ, and allows for comprehensive understanding of regulatory mechanisms that are engaged in precise biological processes based on the structures and functions of different genes [26,27]. With the progression of high-throughput sequencing platforms such as Illumina RNA sequencing, genome-wide expression profiles have been investigated in several species including plants, animals, and microorganisms [28].

In the present study, transcriptome sequencing was applied to identify the ethyl acetate synthesis mechanism. This is the first report of a transcriptomic study of ethyl acetate synthesis in aroma-producing yeasts. The results provide important information for enrichment of genetic resources. The transcriptome analysis was used to generate expression profiles and discover differentially expressed genes (DEGs) among samples during the synthesis of ethyl acetate.

## 2. Materials and Methods

### 2.1. Sampling 

Activated *W. anomalus* strain YF1503 was inoculated into sorghum extract medium. Then, 6% ethanol (*v*/*v*) and 0.01% acetic acid (*v*/*v*) were added to the sorghum extract medium with an inoculation fraction of 4% (*v*/*v*). Fermentation conditions included a temperature of 20 °C and 220 rpm mixing. The experiments were repeated independently three times, and samples were collected at different stages of fermentation (0, 24, 48, 72, 96, and 120 h). The ethyl acetate concentration analyses were conducted with an HPLC (HPLC-1260 Agilent Technologies, Santa Clara, CA, USA) using a ZORBAX Eclipse Plus C−18 column with dimensions 4.6 mm × 150 mm × 5μm (Agilent Technology Co., Ltd.), sampling after the different fermentation periods, which are consistent with the previous reports. The following conditions apply to HPLC detection: Methanol and 0.1 mol/L potassium dihydrogen phosphate buffer were combined in the same proportion as the mobile phase. The column temperature was 35 °C, the flow rate was 1 mL/min, the detection wavelength was 210 nm, the injection volume was 5 L, and the ethyl acetate detection period was 5.0 min. The sampling times were determined according to changes observed in the ethyl acetate concentration in preliminary experiments (Appendix A) [20].

### 2.2. RNA Extraction and Sequencing

Total RNA was extracted from the 18 samples collected at different stages of fermentation (0, 24, 48, 72, 96, and 120 h) following the manufacturer’s guidelines of the RNA Extraction Kit (Tiangen Biotech, Beijing, China). RNA quality was evaluated and high-quality total RNA was used for library construction and sequencing by Shanghai Majorbio Bio-Pharm Technology (Shanghai, China). Poly (A) mRNAs were isolated from 10 μg of total RNA using oligo (dT) beads, then fragmented with a fragmentation buffer and reverse-transcribed to cDNA using random primers. Thereafter, index adapters were ligated to the blunt ends of cDNA molecules. Three biological replicates were processed for each period of ethyl acetate synthesis by YF1503, resulting in a total of 18 cDNA libraries. Each cDNA library was sequenced on an Illumina Hiseq2000 (Illumina, San Diego, CA, USA) platform to create paired-end reads. 

### 2.3. Transcriptome Sequencing Analysis 

The analysis of differentially expressed transcripts from cDNA libraries was performed using a model based on the negative binomial distribution with the DEG sequence [29]. The mRNA abundance of yeast genes was quantified using the fragments per kilobase of exon model per million mapped fragments (FPKM) technique. Transcription factors with FPKM > 0 under the control conditions and at different stages of fermentation were applied to determine fold changes in up- or down-regulated expression [30]. DEGs were also annotated with Gene Ontology (GO) assignments and KEGG pathways. The criteria FDR (False discovery rate) < 0.05 was used to screen DEGs and acted as a threshold for significant GO terms and KEGG annotation. GO enrichment analysis was implemented by the topGO R package using Kolmogorov–Smirnov tests, while significance of the KEGG pathway was tested by KOBAS 3.0 software [31]. 

### 2.4. Transcription Factor Identification and Short Time-Series Expression Miner Analysis

Unigenes that were annotated as transcription factor genes, for instance, bZIP, bHLH, and MYB, were then selected for further analysis. The differentially expressed transcription factor genes and identified temporal expression profiles were evaluated by the Short Time-series Expression Miner V1.3.13 (STEM) program [32]. Gene expression levels at six time points were used, and STEM profiles were clustered, with all parameters set to the default value. A Venn diagram of DEGs and transcription factor genes at different stages of fermentation was generated using http://bioinformatics.psb.ugent.be/cgi-bin/liste/Venn/calculate_venn.htpl (accessed on 2 January 2022).

### 2.5. Expression Analysis of Transcription Factors by Real-Time Quantitative PCR

The relative expressions of transcription factor genes at different stages of fermentation were examined using real-time quantitative PCR (RT-qPCR). For each yeast sample, 5 μg of total RNA were reverse-transcribed to cDNA in a 50 μL reaction using a Prime Script 1st Strand cDNA Synthesis Kit, and then used as a template for RT-qPCR. The primers for RT-qPCR analysis were designed from a non-conserved region preferentially designed from 3′-untranslated regions using Primer-BLAST (http://www.ncbi.nlm.nih.gov/tools/primer-blast/) (accessed on 12 January 2022). RT-qPCR was performed in three biological and technical replicates. The comparative quantitative (ΔΔCT) procedure was implemented to calculate the fold changes in the expression levels of target genes [33]. The *W. anomalus* BS91 actin gene was used as an internal control [34]. Primers were designed in accordance with the sequencing data that are listed in Appendix A.

### 2.6. Statistical Analysis

All the data in this study are presented as the mean and standard error of three independent replicates. The data were analyzed using a Student’s *t*-test with SAS9.3 software (accessed on 24 January 2022).

## 3. Results

### 3.1. Overview of the RNA Sequencing Results

To evaluate global transcriptome changes occurring in *W. anomalus* YF1503, total RNA samples were extracted after different periods (0, 24, 48, 72, 96, and 120 h) of fermentation (Appendix A). From the sequencing results, a total of 46040584, 44570516, 44851995, 43271427, 45610435, and 43967872 raw reads were obtained from six cDNA libraries (Table 1). After removing the short raw reads and quality inspection, the RNA sequence produced 45.3, 43.7, 44, 41.9, 44.2, and 42.7 million clean reads from the six libraries, respectively. All of these reads were used for further de novo assembly. All sequencing data are available through the NCBI Sequence Read Archive under the accession number PRJNA610064. The raw reads of our transcriptome data have been deposited into the NCBI Short Read Archive (SRA, http://www.ncbi.nlm.nih.gov/sra/ (accessed on 29 November 2022).) under accession number SRR11233665- SRR11233682. The sequencing error rate is represented by the Q20 values, which represent the likelihood of error assigned to the recognized bases during base calling: 1%. The Q20 values were 97.91%, 97.94%, 98.03%, 97.96%, 97.92%, and 97.93%, and the GC content percentages were 37.19%, 39.17%, 37.81%, 37.85%, 42.22%, and 38.02%, respectively (Table 1). The length distribution of the transcripts is shown in Appendix A. To validate and annotate the assembled unigenes, we compared all the genes obtained from the transcriptome assembly with those in the six public protein databases (GO, KEGG, COG, NR, Swiss-Prot and Pfam), comprehensively obtained functional information for the genes, and calculated statistics for the annotation of each database. The 13,291 unigenes generated were subjected to BLASTX searches (E-value ≤10^−5^) against public protein databases including NCBI NR, GO, COG, and KEGG (Appendix A). 

### 3.2. Analysis of DEGs in the Synthesis of Ethyl Acetate

DEGs of different fermentation periods were investigated to understand the biological mechanisms. A rigorous comparison at p-adjust < 0.05 and |log2fold change| ≥ 1 for up-regulation or ≤ −1 for down-regulation was performed to identify DEGs for different groups. A total of 2665, 3204, 4741, 5172 and 5255 DEGs were distinct as genes that were considerably enhanced or depleted in one group compared with the 0 h group. Among them, there were 1531, 1724, 2332, 2383, and 2715 up-regulated genes and 1134, 1480, 2409, 2789, and 2540 down-regulated genes from the five comparisons, respectively. We found that the number of up- and down-regulated genes varied among the different fermentation periods. Interestingly, as the fermentation progressed, the number of DEGs increased gradually. The highest number of down-regulated genes (2789) was obtained after 96 h. Among these, 1030 DEGs were common to all five fermentation periods. Besides, there were 385, 200, 173, 867, and 787 DEGs specific to the 24, 48, 72, 96, and 120 h fermentation periods, respectively. The number of DEGs with combinations of two, three, four and five periods are shown in the Venn diagram (Figure 1). 

### 3.3. Identification and Expression Pattern Analysis of Transcription Factors in W. Anomalus Strain YF1503

In total, 307 transcription factors were found in *W. anomalus* strain YF1503 and these could be classified into 48 families. The expression levels of the transcription factor families changed with the length of fermentation. The numbers of transcription factor genes from different families are shown in Figure 2. The eight largest transcription factor families were zf-C2H2, bHLH, bZIP, C3H, GATA, histone-like, HMG, and MYB, which accounted for 62.21% (191/307) of the YF1503 transcription factors (Figure 2). 

Different members may exhibit significant diversity in expression abundance among different fermentation periods to adapt to different processes in the ethyl acetate synthesis. To gain insight into the expression patterns in ethyl acetate synthesis, we investigated the relative expression abundance of all the transcription factor genes in YF1503 using the RNA sequence data. Expression pattern analysis of the 48 transcription factor families (307 transcription factors) was performed using log2 (FPKM). The expression patterns of these transcription factors varied from the different fermentation periods (i.e., 24, 48, 72, 96, and 120 h) compared with 0 h (Figure 3). Among the 307 transcription factor genes, almost all of them showed changes in expression abundance during ethyl acetate synthesis, and most of these were down-regulated. Even within the same family, the expression patterns of its members were not exactly the same. In some cases, family members showed opposite patterns, for example, gene00732 and gene01053 in the C2H2 family, and gene04986 and gene00991 in the bHLH family. As the fermentation progressed, most of the members in families with more members did not show obvious differences in expression, while in families with fewer members, they showed more differences in expression.

At the start of the fermentation, the expression levels of the transcription factor genes could be observed in the heat map. Then, as the fermentation progressed, the expression levels of the transcription factor genes changed at 48 and 72 h. At 96 h, many more genes were significantly up- or down-regulated. At 120 h, the expression levels became irregular and ranged between no expression and high expression. Among the fermentation periods, 21 transcription factors were common in all five comparisons. Meanwhile, there were 7, 3, 2, 12, and 13 transcription factors specific to the 24, 48, 72, 96, and 120 h fermentations, respectively. The number of transcription factors under other combinations of five, four, three, and two periods are shown in the Venn diagram (Appendix A).

### 3.4. Gene Functional Enrichment Analysis of YF1503 Transcription Factors 

To identify functional differences potentially related to ethyl acetate production differentiation after different periods of fermentation, we performed GO terms analysis of all the transcription factors in *W. anomalus* YF1503 on the GO database. The top 20 ranked GO terms are listed in Figure 4A. The response to regulation of glycolytic processes and negative regulation of ribosomal protein gene transcription by RNA polymerase II showed the largest degrees of enrichment, followed by regulation of ribosomal protein gene transcription by RNA polymerase II. In addition, the most abundant functional groups in most of the comparisons were related to RNA biosynthesis and positive regulation of transcription. 

In this study, we used Blast2GO to map all transcription factors involved in the synthesis of ethyl acetate related to the KEGG reference pathway to identify biological pathways in the YF1503 transcriptome data (Figure 4B). The pathways with the most representation by the transcription factors were genetic information processing pathways (29 members) and cellular processes (25 members). These pathways provide a valuable resource for investigating specific processes, functions, and pathways during the synthesis of ethyl acetate.

According to the results, “binding” was the highest GO category in the molecular function ontology with 254 transcription factor genes (Figure 5). The second highest GO categories were “nucleic acid binding transcription” and “catalytic activity”, which included 99 and 43 transcription factor genes, respectively. In cellular component ontology, the highest categories were “the cell” and “cell part” with 190 and 187 transcription factor genes, respectively. In biological process ontology, “cellular process” was the highest category with 167 transcription factor genes, and “metabolic process” was the second highest with 163 transcription factor genes. These results indicated great potential for recognition of novel genes engaged in secondary metabolite synthesis. 

### 3.5. Transcription Factors Involved in the Synthesis of Ethyl Acetate 

Transcription factors are basic regulatory proteins that are vital for gene expression regulation. The GO term “transcription factor activity” was enhanced during ethyl acetate synthesis in the aroma-producing *W. anomalus* YF1503. Transcription factors could bind to cis-regulatory elements in the promoter region of the target gene and act via numerous mechanisms, such as DNA–protein interactions and protein–protein interactions.

Generally, these activators and repressors are related to enormous gene families that are unique to YF1503. Screening of the ethyl acetate transcriptome for transcription factors in the aroma-producing YF1503 transcription factor database identified 307 transcription factors from 48 families. The most important transcription factors were related to the zf-C2H2 (62), MYB protein (24), C3H (23), bZIP (22), and bHLH (20) families. These families are active during ethyl acetate synthesis at different points during fermentation with *W. anomalus* YF1503.

To directly observe the important transcription factors involved in the synthesis of ethyl acetate, we made a clearer analysis of the biological roles of these transcription factors. The Short Time-Series Expression Miner V1.3.13 (STEM) program was used to analyze the differentially expressed transcription factors genes and determine a time expression profile. Then, the 307 transcription factors were divided into four main possible model profiles according to the temporal gene expression patterns, and seven significantly different (*p* < 0.05) patterns of gene expression were recognized (Figure 6 and Appendix A). Among them, Profile 1, which included expression patterns 16, 32 and 70, applied to 67 genes whose expression decreased at 96 h and increased at 120 h. In contrast, Profile 2 included expression patterns 148 and 101 and applied to 20 genes whose expression had increased before 96 h and decreased at 96 h.

### 3.6. RT-qPCR Validation of DEGs 

The varying levels of gene expression for different fermentation periods were evaluated using the FPKM method. RT-qPCR analysis was used to confirm the quality of sequencing and for verification of the expression of *W. anomalus* YF1503 transcription factor genes. 

To verify the expression levels, we selected 12 transcription factors from eight families that could be involved in the synthesis of ethyl acetate. The transcription factors were selected using the ethyl acetate content, heat map, and STEM results. These families and transcription factors were C2H2 (gene00732 and gene09989), HMG (gene01129), bHLH (gene01175 and gene08213), bZIP (gene01273 and gene08309), TFB (gene07385), SRF (gene08565), TFC (gene02199 and gene04039), and TFIIH (gene04885) (Figure 7). 

The results of RT-qPCR showed that the 12 selected transcription factors were up-regulated throughout the fermentation, which indicated that these transcription factors had positive regulatory roles in the synthesis of ethyl acetate. Most of the genes had identical expression patterns to those in the transcriptomic data. For example, the expression levels of TFB (gene 07385) and TFC (gene 02199) were relatively high, increasing at 96 h and then beginning to decline at 120 h. The data from this research could be used as a tool for analysis of *W. anomalus* YF1503 transcription factor genes that are differentially expressed as the fermentation progresses. 

## 4. Discussion

Even in non-model organisms without genomic information, transcriptome sequencing is a high-throughput and cost-effective way to generate genetic resources. With technological developments, next-generation sequencing can produce data with 98% accuracy and allow for efficient transcriptome de novo assembly by combining multiple bioinformatics methods [24]. RNA sequencing analysis based on Illumina sequencing technology provides comprehensive and systematic information about gene expression for a wide range of research fields [24]. It can accurately detect changes in the transcript abundance with time. In our previous study, we found that abnormal *W. anomalus* strain YF1503 could produce a high concentration of ethyl acetate under optimum fermentation conditions [20]. Ethyl acetate is the key compound that affects and determines the flavor and quality of *baijiu*. To date, ethyl acetate synthesis has mainly focused on the acquisition of enzyme genes [35]. Previous studies have shown that the synthesis of ethyl acetate in yeast is accomplished by three enzymes: alcohol acetyltransferase, hemiacetal dehydrogenases, and esterases [16,21,36]. However, additional important factors, especially transcription factors, are still largely unknown. Here, we provide the first report on the analysis of *W. anomalus* from the transcriptome level after fermentation, which may be useful for identifying key factors involved in ethyl acetate synthesis.

In this study, RNA sequencing was performed using Illumina sequencing after removing short raw reads and quality inspection. This generated approximately 45.3, 43.7, 44, 41.9, 44.2, and 42.7 million clean reads for the six libraries. Eighteen biological samples of abnormal Hanson’s yeast collected after six fermentation periods were sequenced by the Illumina sequencing platform. Compared with 0 h, 2665, 3204, 4741, 5172, and 5255 DEGs were obtained. Venn diagram analysis showed that 1031 of the DEGs were common throughout the fermentation.

Ethyl acetate synthesis is a complex process where one molecule of acetic acid and one molecule of ethanol react with an esterifying enzyme [37]. In *baijiu* brewing, many esters, including ethyl acetate, are produced by various enzymes from microorganisms that can catalyze biochemical reactions of different substrates [16]. Enzymes are closely related to flavor substances in *baijiu*. There are two main ways to synthesize ethyl acetate: one way is from acid and alcohol under the action of lipase/esterase, and the other way is to generate acetyl coenzyme A and ethanol under the action of alcohol acyltransferase [10,21,35]. Research on the second pathway has been more in-depth, and the mechanism of action is clearer than that in the first pathway. 

Currently, alcohol acyltransferases, which use alcohol and acyl-CoA as substrates to create esters, are the principal method by which yeasts produce ethyl acetate. So, in order to search for genes encoding alcohol acyltransferases in the *W. anomalus* strain YF1503, we paired this with functional annotation analysis. We quantified the levels of alcohol acyltransferase expression in *W. anomalus* strain YF1503 at different phases of fermentation (Appendix A). A good indication that the differential gene expression is directly connected to the synthesis of ethyl acetate is that, when combined with changes in ethyl acetate content, we discovered that the number of DEGs had a similar trend to the ethyl acetate content. The gene expression change was in the front and the product response was in the back. In comparison to other alcohol acyltransferase genes, gene00276 and gene09517 were significantly more up-regulated.

However, in addition to enzymes, transcription factors also play important roles in the catalytic synthesis of ethyl acetate and can affect this process in *baijiu* brewing. As the main switch in complex regulatory networks, transcription factors play an important role in many species by specifically binding with upstream cis-regulatory elements to activate or inhibit the transcription of downstream target genes [23].

Many microorganisms (e.g., fungi, yeast, bacteria, and mold) with unique functions are involved in *baijiu* fermentation [38,39]. Some of the fungi and bacteria can synthesize ethyl acetate only in low yields as flavor components in the fermented product. Among the microorganisms with esterification abilities, yeast has a strong ethyl acetate production capacity [40,41]. Many yeast strains, including *W. anomalus*, can synthesize ethyl acetate in high yields [10]. The strains with high yields of ethyl acetate have been studied in China in liquor products such as Daqu and fermented grains.

From the sequencing data results, we identified 307 transcription factors that were divided into 48 families. During fermentation, genes with the same expression patterns may play important roles in the reaction. There were 21 transcription factors that were common to the five fermentation periods. The number of transcription factors that were common to five, four, three, and two fermentation periods are shown in a Venn diagram (Appendix A). The expression levels of the transcription factor genes for the different fermentation periods showed that the total number of up-regulated genes was less than the number of down-regulated genes. However, as the fermentation time increased, the total expression levels of all transcription factor genes (up-regulated and down-regulated) increased. More of transcription factor genes at 96 and 120 h were down-regulated than up-regulated. The expression levels of transcription factor genes in different families showed that more genes were down-regulated than up-regulated in the initial period of the fermentation, but this trend reversed as the fermentation progressed and more genes were up-regulated than down-regulated. Then, after 72 h, the down-regulated genes outnumbered the up-regulated genes again. In summary, the expression levels of the different families showed that the down-regulated genes outnumbered the up-regulated genes at the start and end of the fermentation, but this trend was reversed in the middle of the fermentation. In summary, we found that the expression levels of most transcription factors turned around at 72–96 h of incubation time, which, combined with the results of GO, KEGG and STEM, was consistent with the results of ethyl acetate production of *W. anomalus* strain YF1503. In other species under stress or other treatment conditions, many transcription factors are induced and play critical roles in adaption to the environment [40]. However, there is still a lack of study on transcription factors in *W. anomalus*. To clarify the specific biological process of these transcription factors, we classified selected transcription factors, including those in GO and KEGG. We aimed to identify transcription factors that play important roles in the synthesis of ethyl acetate through these methods.

In the subcategories, “binding” was the highest GO category in the molecular function ontology with 254 transcription factor genes, followed by “nucleic acid binding transcription” with 99 transcription factor genes. Many differentially expressed transcription factor genes are involved in ethyl acetate synthesis, and these transcription factors are also enriched in different pathways. In this study, the most abundant KEGG pathways were genetic information processing and cellular processes. These transcription factors were also enriched in different pathways, including the autophagy pathway and Hippo signaling pathway. In many species, the research on these pathways has been very in-depth. They are relatively common in cells and have very important functions. However, the relationship between these pathways and ethyl acetate synthesis needs further verification.

To date, no studies on transcription factors in ester synthesis have been reported. In this study, based on the analysis of transcriptome sequencing, including Heat map and STEM, we conducted RT-qPCR analysis of some of the transcription factors with obvious differences in gene expression for the different fermentation periods. The RT-qPCR analysis validated the transcriptome data, which indicated that the data obtained in this research could be a useful resource for analysis of key factors in ethyl acetate synthesis.

Different amounts of ethyl acetate are synthesized at various periods as a result of variations in the transcript levels of genes-encoding ethyl acetate synthase-related genes. In actuality, multiple related transcription factors also control the genes involved in ethyl acetate synthase activity. Seven main expression patterns—Proile16, Proile32, Proile70, Proile148, Proile101, Proile0, and Proile41—were analyzed based on the results of the STEM (V1.3.13) expression analysis software. The expression patterns of transcription factors in Proile148 and Proile101 matched those of the alcohol acyltransferase-encoding genes 00276 and 09517 (Appendix A). For instance, in Proile101, the expression level of transcription factor gene11140 (a member of the C3H family of transcription factors) grew progressively over the course of fermentation, decreased somewhat at 96 h, and then increased once again at 120 h. Comparable to gene00276 and gene09517, transcription factors gene00991 and gene01592 from the bHLH family, as well as gene03761, gene08969, and gene11176 from the GATA family, all showed similar patterns of expression in Proile148. When ethanol and acetic acid are present in the system, it is possible to predict that specific transcription factor family members within *W. anomalus* strain YF1503 catalyze the production of ethyl acetate by controlling the expression of the alcohol acyltransferase genes, and that the two should work in concert. Despite the fact that the molecular mechanism of transcription factor regulation of ethyl acetate synthesis has not been reported, this study enables us to make educated predictions about the potential transcription factors involved and their primary regulatory enzymes as well as other key genes, laying the groundwork for a more thorough understanding of the molecular mechanism of ethyl acetate.

To further evaluate the molecular mechanism of ethyl acetate synthesis, other biological experiments are needed to prove the specific functions of these candidate transcription factors. For example, CRISPR technology could be used to knock out important transcription factors and then verify the functions of mutants. Alternatively, well-known yeast two-hybrid technology could be used to screen yeast libraries, and screen for important proteins that interact with the transcription factors. The results could be used to clarify the molecular mechanism of YF1503′s high yield of ethyl acetate.

## 5. Conclusions

Three hundred and seven transcription factors were found in *baijiu* strain YF1503 and classified into 48 families using high-throughput RNA sequencing. Their expression patterns showed dynamic changes after different fermentation periods. This study is the first to report a dynamic and systematic investigation of TFs involved in ethyl acetate synthesis. A total of 21 TFs were common to the five fermentation periods, and these may be suitable candidates for further study. We conducted RT-qPCR analysis of some of the novel transcription factor genes identified based on the analysis of transcriptome sequencing, including Heat map and STEM analyses, and we found that they have potential for use as target genes to regulate ethyl acetate synthesis. The results for transcription factor s in this study provide an important foundation for understanding molecular regulation of ethyl acetate synthesis in *W. anomalus* strain YF1503.

## Figures and Tables

**Figure 1 genes-13-02341-f001:**
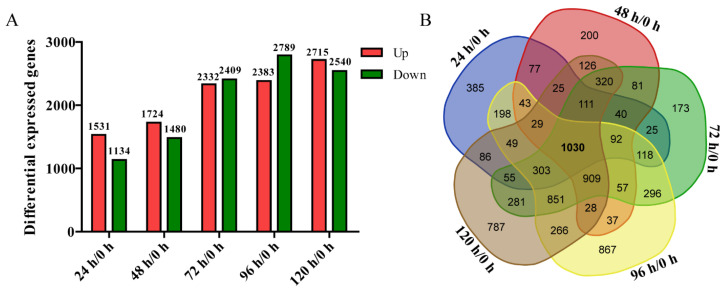
Summary of differentially expressed genes (DEGs) in *W. anomalus* YF1503 and Venn diagram of DGEs under different fermentation times. (**A**) The number of up-regulated (Up) or down-regulated (Down) genes under different fermentation times. (**B**) Venn diagrams showing the total number of significantly DEGs (*p*-value ≤ 0.05) at 24 h, 48 h, 72 h, 96 h, and 120 h compared with 0 h.

**Figure 2 genes-13-02341-f002:**
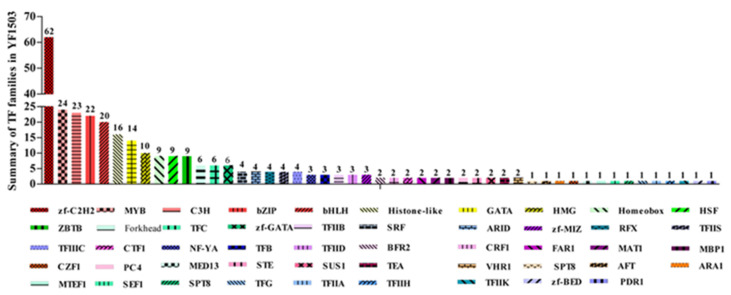
Summary of the transcription factor genes in *W. anomalus* strain YF1503. The number of up-regulated (Up) or down-regulated (Down) genes for different fermentation periods.

**Figure 3 genes-13-02341-f003:**
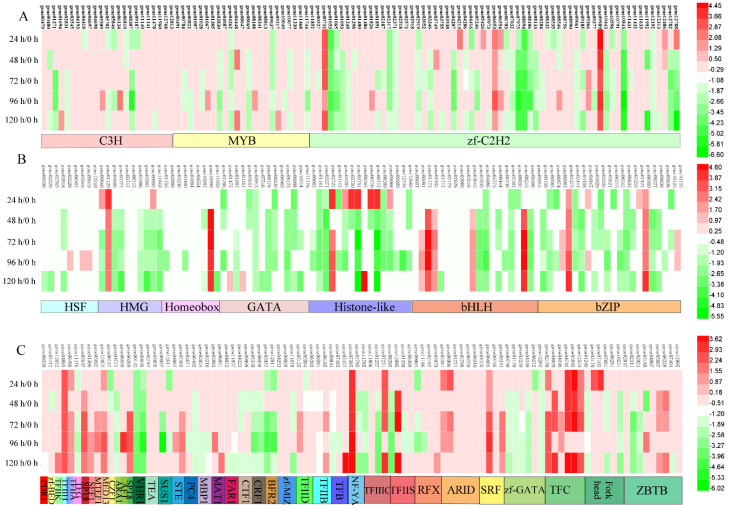
Heat map of expression profiles (in log2−based FPKM) of TF genes in yeast YF1503. The expression abundance of each transcript is represented by the color of the bar, with green indicating a low level and red indicating a high level. (**A**) The expression profiles of zf-C2H2, MYB and C3H family TF genes. (**B**) The expression profiles of HSF, HMG, Homeobox, GATA, Histone-like, bHLH and bZIP family TF genes. (**C**) The expression profiles of PDR1, zf-BED, TFIIK, TFIIH, TFIIA, TFG, SPT8, SEF1, MTEF1, MED13, CZF1, ARA1, AFT, VHR1, TEA, SUS1, STE, PC4, MBP1, MAT1, FAR1, CTF1, CRF1, BFR2, zf-MIZ, TFIID, TFIIB, TFB, NF-YA, TFIIIC, TFIIS, RFX, ARID, SRF, zf-GATA, TFC, Fork head and ZBTB family TF genes.

**Figure 4 genes-13-02341-f004:**
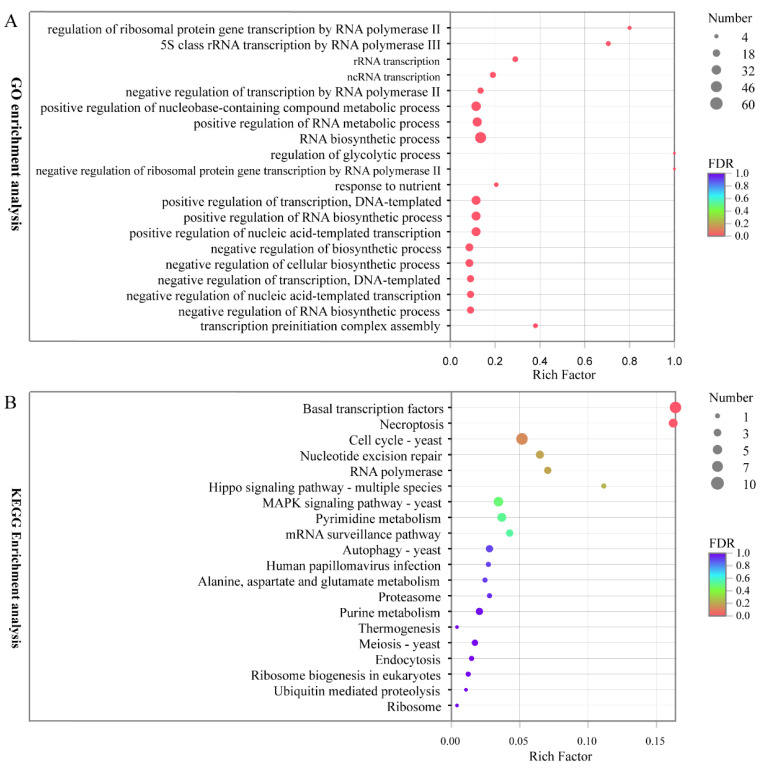
GO and KEGG enrichments of TFs. (**A**) GO enrichment analysis. (**B**) KEGG enrichment analysis. The *x*-axis shows the enrichment factor. Blue represents a high q value and red represents a low q value.

**Figure 5 genes-13-02341-f005:**
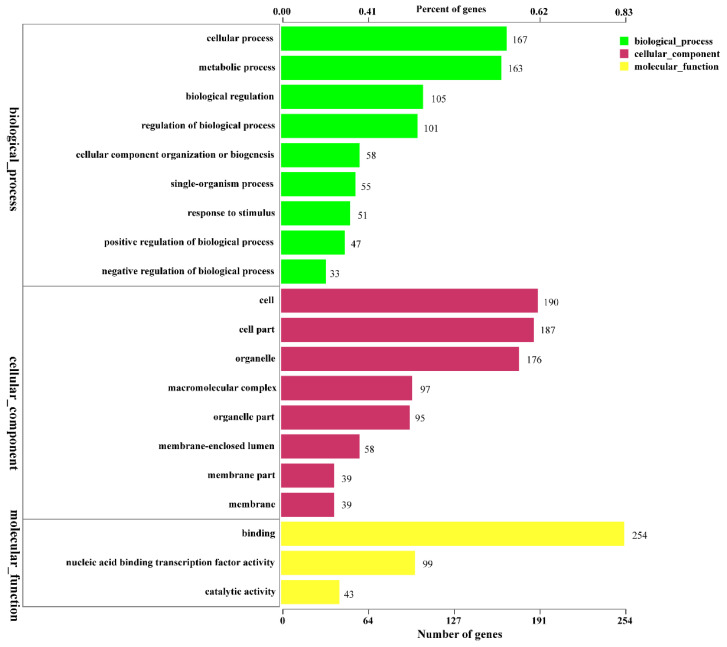
Classification of GO annotations in yeast YF1503. The *x*-axis shows the number of genes.

**Figure 6 genes-13-02341-f006:**
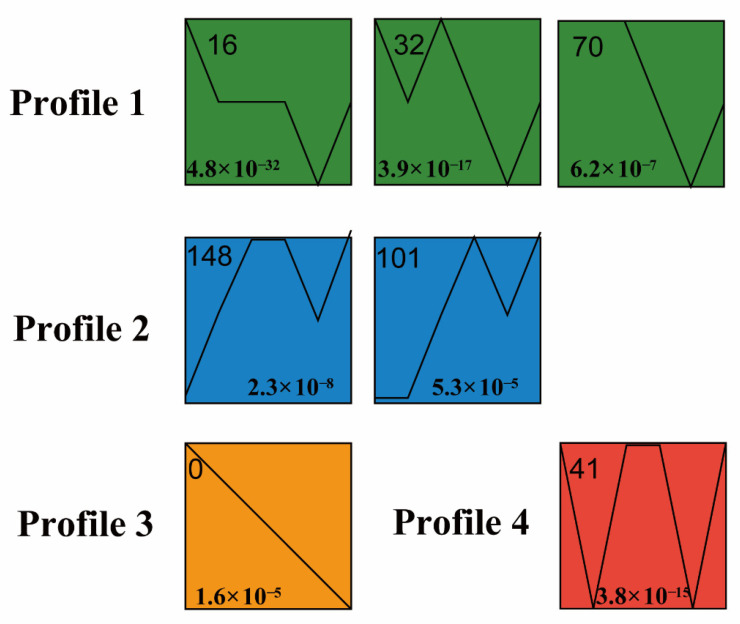
Seven main possible model profiles recognized by STEM. The Short Time-Series Expression Miner (STEM) program classified the 307 TF genes into four main possible model profiles according to the temporal gene expression patterns, and within these recognized 7 significant patterns of gene expression (*p* < 0.05). The broken line in the figure shows the trend of expression quantity changing with time, and the value at the lower left corner represents the corresponding significant level (*p*−value).

**Figure 7 genes-13-02341-f007:**
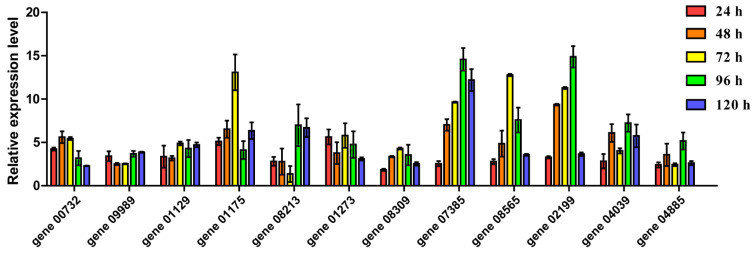
Expression analysis of 12 TF genes at different fermentation times using qRT-PCR. The experiments were repeated three times.

**Table 1 genes-13-02341-t001:** Summary of basic statistics for the *W. anomalus* YF1503 transcriptome sequencing data.

Sample	Raw Reads	Clean Reads	Clean Data Rate (%)	Q20(%)	GC Content(%)
0 h	46,040,584	45,382,701	98.57	97.91	37.19
24 h	44,570,516	43,779,746	98.23	97.94	39.17
48 h	44,851,995	44,058,827	98.23	98.03	37.81
72 h	43,271,427	41,956,821	96.96	97.96	37.85
96 h	45,610,435	44,290,539	97.11	97.92	42.22
120 h	43,967,872	42,710,191	97.14	97.93	38.02

## Data Availability

All raw data used during the study are available from the corresponding author by request.

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
