# Peer review of "Dynamic Transcriptome Analysis Reveals Transcription Factors Involved in the Synthesis of Ethyl Acetate in Aroma-Producing Yeast"

_genes, 2022, doi:10.3390/genes13122341_

Round 1

Reviewer 1 Report

Genes-2037973

1.       The content of the paper entitled “Dynamic transcriptome analysis reveals transcription factors involved in the synthesis of ethyl acetate in aroma-producing yeast” is a unique and worthy research topic to work upon.

2.       The abstract is informative, and it reflects the body of the paper.

3.       The introduction provides sufficient background information for readers in the immediate field to understand the problem/hypotheses.

4.       Line 76: What is the condition of “three experiments…”.? please clarify and add the information.

5.       Please added the condition of HPLC.

6.       What is the Q20 value? Please give a specific name.

7.       Line 127: Please add the strain of W. anomalus.

8.       Line 180: change Fig.2 to Figure 2.

9.       Discussion can be improved by providing mechanisms involved in the changes.

Reviewer 2 Report

This is an interesting paper related to the possible identification of transcription factors involved in the synthesis of ethyl acetate in aroma-producing yeast. However since Baijiu is a traditional Chinese alcoholic beverage there are some aspects of the beverage production that needs to be clarified as:

1. Explain why and how the W. anomalous was activated. What activated mean?

2, Why the authors added ethanol and acetic acid to the fermentation?

In qPCR methodology why only actin was used as internal control? Were other genes tested?

The TFs identified as DEGs could be deeply discussed related to what genes can be activated or repressed by them and their relationship with the genes involved in the synthesis of ethyl acetate. Since this is the main aim of the work these points need to be deeply discussed

Round 2

Reviewer 2 Report

The manuscript was improved and new results were added. The discussion was expanded explaining several aspects of the TFs expression. This reviewer suggests that the authors introduce some elements about the Baijiu production since it is a traditional Chinese alcoholic beverage. A revision in the name of genes (italic), proteins (non-italic), and some misspellings needs to be improved.